



# 1    On the Freezing Time of Supercooled Drops in Developing

# 2    Convective Clouds

**Jing Yang[1], Zhien Wang[1], and Andrew Heymsfield[2]**
[1] {Department of Atmospheric Science, University of Wyoming, Laramie, WY}
[2] {National Center for Atmospheric Research, Boulder, CO}
Correspondence to: Zhien Wang (zwang@uwyo.edu)
**Abstract**
In this study, the particle size distributions (PSDs) measured in fresh developing maritime
convective clouds sampled during the Ice in Clouds-Tropical (ICE-T) project are shown and
compared with the PSDs modeled using a parcel model containing a spectral bin microphysics
scheme. The observations suggest that the "first ice" in convective clouds is small. To interpret
the observed ice PSDs, the freezing times and temperatures of supercooled drops are analyzed.
The results indicate that the freezing time is longer for large drops than it is for small drops. Due



to instrumental limitations, freezing drops cannot be identified until they exhibit obvious shape
deformation. If the updraft is strong enough, large freezing drops can be carried upwards to a
lower temperature than their nucleation temperature before obvious shape deformation occurs. In
models, drop freezing is assumed to be instantaneous, which is not realistic; thus, the model
yields a broader "first ice" PSD than is observed. This study allows us to interpret the observed
ice PSDs in fresh developing convective clouds from the perspective of the freezing time of
supercooled drops and notes the deficiency of instantaneous drop freezing in models. To better
understand the mechanisms of drop freezing and ice initiation in convective clouds, more
laboratory experiments and in situ measurements are needed in the future.
**1.      Introduction**
Ice initiation in convective clouds is still not well understood (Cantrell and Heymsfield, 2005;
Lawson et al., 2015; Yang et al., 2016), and it remains one of the main sources of uncertainties in
numerical models (Khain et al., 2015). Observations suggest that ice initiation in convective
clouds is strongly related to the freezing of supercooled drops (Rangno and Hobbs, 2005;
Lawson et al., 2015; Yang et al., 2016; Field et al., 2017). Supercooled drops do not fully freeze
instantaneously, and during airborne measurements, freezing drops cannot be observed until they
have experienced obvious deformation. The freezing rate of supercooled drops depends on the
rate of heat transfer between the drop and ambient air (Pruppacher and Klett, 1997). Typically,
the freezing process comprises four stages (Hindmarch et al., 2003): 1) the supercooling stage,
during which a drop is supercooled to its nucleation temperature; 2) the recalescence stage,
during which rapid kinetic ice nucleation occurs, which results in a rapid drop in temperature that
is terminated when the drop temperature reaches 0 °C; 3) the freezing stage, during which the
liquid part of a drop continuously freezes and the drop temperature remains at 0 °C; and 4) the



cooling stage, during which the frozen drop cools to the ambient temperature.
A number of laboratory experiments have been performed to study the freezing of supercooled
drops. For example, Johnson and Hallett (1968) showed that the freezing time of supercooled
drops decreases with decreasing ambient temperature. In typical air conditions, it takes
approximately 400 s for a stationary millimeter-sized drop to completely freeze at -5 °C under a
pressure of 1 atm; this freezing time is reduced to approximately 200 s at -10 °C. They also
showed that the freezing rate of supercooled drops is related to the composition of air and that
the freezing time of a millimeter-sized drop in helium and hydrogen is only one-fifth of that in
air. Hindmarsh et al. (2003) showed that the freezing time increases with increasing drop size. In
addition, Hindmarsh et al. (2003) used experimental results to discuss the accuracy of three drop
freezing models: the uniform temperature model, the inward freezing model and the outward
freezing model. All three of these models have fairly good accuracy in modeling drop
temperatures and freezing times, and there are only minor differences between them.
In most numerical weather prediction models (NWPMs) and global climate models (GCMs), the
freezing of supercooled drops is assumed to be instantaneous, because it is difficult to track the
freezing stage of every particle in models and because there are no good observations with which
to evaluate the modeled ice microphysics in detail. Phillips et al. (2015) implemented time-
dependent freezing for raindrops in a cloud model using spectral bin microphysics (SBM). Their
sensitivity tests showed that time-dependent drop freezing delays the formation of hail in
convective clouds; however, their model was unable to track the freezing stage of every particle.
Using a simplified cloud parcel model and an electromagnetic scattering model, Kumjian et al.
(2012) showed that the modeled radar polarimetric variables for convective clouds are more
consistent with observations if time-dependent drop freezing is considered. However, drop



freezing in fresh developing convective clouds has rarely been discussed. Thus, to better
understand ice initiation in convective clouds and to evaluate the modeled microphysics, more
observations are needed.
Aircraft in situ measurements are necessary to improve our current understanding of ice
initiation in convective clouds and to evaluate model simulations. Traditional in situ
measurements can rarely identify ice that is smaller than 200 $\mu$m in diameter. The 3-View Cloud
Particle Imager (3V-CPI) is a good tool with which to record small particle images, and it can be
used to identify small ice (Field et al., 2017). During the Ice in Clouds-Topical (ICE-T) project,
the 3V-CPI that was operated on the SPEC Learjet yielded high-resolution particle images and
particle size distributions (PSDs). The 3V-CPI measurements suggest that the observed "first
ice" in fresh developing convective clouds are all small ice (Lawson et al., 2015); however, the
results of some other studies have suggested that larger supercooled drops may freeze before
smaller drops (Bigg, 1953; Heymsfield, 2013). This raises the question: why is the observed
"first ice" in convective updrafts small? Understanding the freezing time of supercooled drops is
helpful for interpreting the observed ice PSDs in developing convective clouds. In addition,
determining the size of "first ice" is important for understanding secondary ice generation
process(es). This study aims to analyze the observed PSDs in developing convective clouds
using the data collected during the ICE-T project, as well as to interpret these observations
through the perspective of the freezing time of supercooled drops. This paper is organized as
follows: Section 2 introduces the dataset and the analytical method; Section 3 discusses the
results; and a summary is given in Section 4.
**2.      Dataset and Analysis Method**



## 2.1 Calculation of the freezing time of supercooled drops


The calculation of the freezing time and temperature of supercooled drops is governed by a
series of heat transfer and phase change equations. These detailed equations have been described
in previous studies (e.g., Dye and Hobbs et al., 1968; Hindmarsh et al., 2003). The drop
temperature change is balanced by convective heat transfer (i.e., ventilation), radiation and latent
heat terms. In this calculation, a supercooled drop is assumed to be carried upward by an updraft,
which ascends adiabatically. The terminal velocity of the drop follows that defined by Foote and
Du Toit (1969). In this calculation, diffusional growth is included but coalescence is neglected.
The initial drop temperature is the same as the ambient air temperature. The temperature inside
the drop is assumed to be uniform; this is a reasonable assumption because water and ice have a
larger thermal conductivity than air and because of the internal mixing of liquid within the drop
(Yao and Schrock, 1976). Hindmarsh et al. (2003) showed that including temperature variations
inside the drop has a minor impact on the results. The freezing time is defined as the time period
from the start to the end of the drop freezing.

## 2.2 Aircraft measurements during ICE-T


The ICE-T project was conducted in July 2011 over the Caribbean Sea, near the U.S. Virgin
Islands; its goal was to study ice generation in tropical maritime convective clouds. Both the
National Center for Atmospheric Research (NCAR) C-130 aircraft and the SPEC Incorporated
Learjet were deployed during ICE-T.
The SPEC Learjet was equipped with various instruments that were used to study the
microphysics in convective clouds during ICE-T. The primary goal of the Learjet was to make
repeated penetrations in fresh developing convective updrafts near the cloud top. These



instruments included a fast forward-scattering spectrometer probe (FFSSP); a CPI; a two-
dimensional stereo (2D-S) probe; a high-volume precipitation spectrometer (HVPS-3), and a
Rosemount temperature probe. The measurements obtained using the FFSSP, CPI, 2D-S, and
HVPS were combined to generate the PSDs. CPI images were used to identify liquid drops and
ice particles that were smaller than 500 $\mu$m in diameter, and these percentages of drops and ice
particles were applied to the 2D-S PSDs. The 2D-S and HVPS images were used to identify
drops and ice particles that were larger than 500 $\mu$m in diameter. More information about the
processing of the Learjet data can be found in Lawson et al. (2015).
The NCAR C-130 was not used to repeatedly penetrate fresh developing convective clouds
during ICE-T; instead, it penetrated convective clouds at different stages of their development.
Most of these penetrations occurred far below the cloud top, although some were near the cloud
top (Heymsfield et al. 2014). The instruments used here included an FFSSP, a two-dimensional
cloud (2D-C) probe, a two-dimensional precipitation (2D-P) probe, and a Rosemount
temperature probe. The Wyoming Cloud Radar (WCR; Wang et al. 2012) was operated on the C-
130 to obtain 2D reflectivity structures, and the Wyoming Cloud Lidar (WCL; Wang et al. 2009)
was used to identify liquid-dominated and ice-dominated clouds.
**2.3      Parcel model simulation**
In this study, we compare the PSDs modeled using a parcel model containing SBM to those
observed on the aircraft. The SBM was developed by Hebrew University (Khain et al. 2000) and
has been implemented in the Weather Forecast and Research model (WRF; Lynn et al. 2005).
Time-dependent drop freezing is not included in this scheme. The purpose of this simulation is
not to evaluate the modeled results using observations, but instead to reveal the deficiency of



instantaneous drop freezing in SBM and its inability to capture the observed rapid ice generation.
The modeled parcel has a depth of 500 m. The observed drop size distribution at -6 °C is used as
an input. The vertical air velocity is 10 m/s, which is a typical mean updraft strength in the
convective clouds sampled during ICE-T. The hydrometeor types include cloud drop/rain,
ice/snow, and graupel; the PSD of each hydrometeor type has 33 mass bins. The ice nucleation
mechanisms include immersion freezing using Bigg's parameterization (1953),
deposition/condensation nucleation (Meyer et al., 1992), contact nucleation (Meyer et al., 1992),
and the Hallett-Mossop process (Hallett and Mossop, 1974). Other ice microphysics processes
include riming, coalescence and diffusional growth. During every time step, 1% of the aerosols
in the ambient air are assumed to become entrained into the cloud parcel. The ambient aerosol
size distribution is observed using a high-flow dual-channel differential mobility analyzer
(HDDMA; DeMott et al., 2016) and a Passive Cavity Aerosol Spectrometer Probe (PCASP;
Baumgardner et al., 2011) operated on the C-130.
**3.    Results and Discussion**
**3.1    Comparison of observed and modeled particle size distributions**
Fig. 1 shows the size distributions measured by the Learjet and those modeled using a parcel
model with SBM. The simulation data on the left panels include all of the ice physics
implemented in the SBM, while liquid-ice collision is turned off for the right panels. The Learjet
measurements suggest that the ice particles observed in fresh developing convective clouds are
relatively small (20-300 $\mu$m in diameter) between -7 °C and -10 °C and that the PSD of ice
broadens as the temperature decreases. The modeled ice PSD is much broader than that observed
between -7 °C and -10 °C. The deposition/condensation nucleation exhibits the largest



contribution to the modeled ice PSDs (Fig. 1d). Immersion freezing, contact freezing and the
Hallett-Mossop process contribute less to the modeled ice PSDs. Small ice particles are mostly
formed by deposition/condensation nucleation, whereas large ice is produced by immersion
freezing and drop-ice collision (Fig. 1d and h).
An obvious difference between the observed and modeled ice PSDs is that large ice is not
observed between -7 °C and -10 °C but is found in the modeled results (Fig. 1d). There are three
possible explanations for this: first, large freezing (or frozen) drops cannot be identified from the
images taken by the probes, or the sampling volume of the probes is too small; second, the
modeled results are not realistic; third, there could be a combination of the first and second
possibilities. Previous studies have suggested that during immersion freezing, large drops have a
higher probability of freezing than small drops at the same temperature (Bigg, 1953). In addition,
small ice that is generated by other mechanisms (e.g., deposition/condensation nucleation,
secondary ice) can be quickly collected by large drops in convective clouds, which results in the
freezing of large drops. There is no evidence that large drops do not freeze between -7 °C and -
10 °C. In the observations, only non-spherical particles are regarded as ice, but freezing drops
exhibit no (obvious) shape deformation during the early stage of freezing (Johnson and Hallett,
1968; Hindmarsh et al., 2003). Due to the limitations of the instruments, freezing drops that do
not exhibit obvious shape deformation cannot be identified; thus, the first possibility may apply.
On the other hand, in the model simulations, drop freezing is assumed to be instantaneous, which
could result in a broad ice PSD at warm temperatures; because this is not true in natural clouds,
the second possibility may also apply. Therefore, the large difference between the measured and
simulated ice PSDs is probably both observation- and model-related.
Examples of particle images collected by the 2D-C on the C-130 and the CPI on the Learjet are



shown in Fig. 2. Both the 2D-C and CPI images were measured near the cloud top in the updraft
cores of developing convective clouds. As noted in the figure, the observed ice particles mostly
comprise small frozen drops between -8 °C and -10 °C (Fig. 2c). Some particles may have just
begun freezing because they exhibit slight shape deformation, as shown by the particle images in
the red box in Fig. 2c; however, we have no other evidence with which to confirm this. Between
-10 °C and -13 °C, we observe more ice particles, including both large and small frozen drops, as
well as rimed graupels (Fig. 2a and b). Columns and plates were also observed. Considering the
time that is needed for columns and plates to grow, they were probably generated at a warmer
temperature than is observed. Due to the relatively low resolution of the 2D-S, 2D-C, HVPS and
2D-P images, large freezing (or frozen) drops that exhibit no obvious shape deformation cannot
be identified, and they are thus regarded as drops. In some spherical CPI particle images, it is
also difficult to determine whether the particles have begun freezing or not, because freezing
drops exhibit no (or no obvious) shape deformation during the early stages of freezing (e.g.,
Johnson and Hallett, 1968; Hindmarsh et al., 2003).
**3.2     Freezing time of supercooled drops**
To better understand the observed PSDs, we analyze the freezing time of supercooled drops in
this section. Fig. 3 shows the changes in drop temperature and ice mass fraction with changes in
time and ambient temperature. The updraft velocity is assumed to be 10 m/s. Drops and air
parcels ascend from -6 °C (~520 mb, ~5600 m). The nucleation temperature, which is the
temperature at which drops begin to freeze, is assumed to be -8 °C. The figure demonstrates that
a drop with a radius of 100 $\mu$m cools from -6 °C to -8 °C and begins to freeze at approximately
23 s. The latent heat released due to freezing leads to a sudden drop in temperature from -8 °C to
0 °C (Fig. 3a), and the ice mass fraction increases from 0 to 0.1 (Fig. 3b). It takes approximately



4 seconds for the drop to fully freeze; during freezing, the drop temperature remains at 0 °C (Fig.
3a), and the ice mass fraction continuously increases (Fig. 3b). After completely freezing, the
frozen drop rapidly cools due to the large difference between the ambient temperature and the
drop surface temperature. The cooling rate slows down when the frozen drop temperature
approaches the ambient temperature. According to its equations, the cooling rate for a drop in the
updraft is largely controlled by convective heat transfer, rather than radiation or diffusional
growth. If significant riming occurs on the freezing (frozen) drop surface, the cooling rate could
be slower, and the freezing time could thus be longer due to the latent heat release that occurs
during riming (Phillips et al., 2015). The drop temperature changes in a similar way for larger
drops as it does for small drops. However, due to their higher terminal velocity, it takes longer
for larger drops to reach their nucleation temperature (-8 °C). Drops with radii of 250 $\mu$m and
500 $\mu$m begin to freeze at 28 s and 43 s, respectively (Fig. 3a), and their ambient temperatures
are approximately -8.1 °C and -8.15 °C (Fig. 3c), respectively. In addition, it takes longer for
larger drops to completely freeze. Drops with radii of 250 $\mu$m and 500 $\mu$m require approximately
15 s and 35 s, respectively, to fully freeze (Fig. 3a); these frozen drops are found at temperatures
of -9.2 °C and -9.95 °C, respectively (Fig. 3c).
Fig. 4 shows the freezing time and frozen temperature as functions of the drop radius for
different vertical air velocities and nucleation temperatures. The freezing time represents the
time period from the start of drop freezing to the end of drop freezing. The figure shows that the
freezing time increases as the radius increases. For the same nucleation temperature, drops freeze
faster in stronger updrafts than they do in weaker ones (Fig. 4a); however, their frozen
temperatures are colder in stronger updrafts (Fig. 4b). In addition, for the same updraft strength,
a drop freezes faster when its nucleation temperature is lower, and it fully freezes at colder





temperatures. Moreover, for the same drop radius, the effect of the updraft strength on the
freezing time is smaller if a drop nucleates at a lower temperature, as is indicated by the smaller
differences between the solid, dashed and dotted lines for colder nucleation temperatures (Fig.
4a); however, its impact on frozen temperature does not vary substantially with different
nucleation temperatures (Fig. 4b).
Large drops may begin to freeze at warmer temperatures than small drops (Bigg, 1953). Fig. 5
shows the nucleation temperature and frozen temperature as functions of the drop radius. The
nucleation temperature is the temperature at which drops have a $10^{-4}\%$ probability of freezing, as
determined based on Bigg's parametrization for immersion freezing. This probability is low
because of the low concentration of immersion ice nuclei that are present at warm temperatures.
The figure shows that large drops may begin to freeze at warmer temperatures than small drops;
however, due to their longer freezing times, large drops may fully freeze at colder temperatures
than small drops if the updraft is strong enough. Immersion freezing is not the only ice
nucleation mechanism. In convective clouds, small ice can be generated at warmer temperatures
by other mechanisms (e.g., condensation/deposition nucleation). The ice PSD measured by the
Learjet indicates that large frozen drops were observed at colder temperatures than small ice, but
it is not known whether these large drops started to freeze before or after the small droplets, and
the mechanisms that lead to drop freezing are not well understood.
**3.3    Discussion**
The above analysis indicates that large frozen drops are observed at relatively colder
temperatures than small ice in strong updrafts of convective clouds but that they may begin to
freeze at warmer temperatures. If the vertical air velocity is not strong enough, large drops may



descend or remain at the same level for long periods of time, and they may freeze if their
temperature reaches the nucleation temperature. An example of this is shown in Fig. 6. In this
case, penetration occurred approximately 500 m below the cloud top, as is indicated by the WCR
reflectivity (Fig. 6a). The WCL power (Fig. 6c) quickly attenuated and the WCL depolarization
ratio (Fig. 6d) is relatively low, which indicates that this cloud was dominated by liquid drops.
At the flight level, the temperature (Fig. 6e) ranges from -4 °C to -4.5 °C in the updraft and is
approximately -5 °C near the cloud edge. The maximum updraft velocity is 7 m/s, and the mean
updraft velocity is approximately 3 m/s. The Doppler velocity (Fig. 6b) is negative in most areas
of the clouds, and its maximum value is approximately 4 m/s. The 2D-C images clearly show the
existence of ice (Fig. 6f). Most of the ice particles are frozen drops and graupel, and some are
needles and columns. The graupel may fall from above; thus, they may start freezing at a colder
temperature than the flight level temperature. Considering the time that is needed for the drops to
freeze and for the needles and columns to grow through vapor diffusion, this ice may have
nucleated when the cloud top was lower than observed.
The freezing of supercooled drops may be associated with some corresponding processes. For
example, drops may break up or shatter during freezing, which can produce multiple ice
fragments and splinters. Mason and Maybank (1960) showed that the freezing of a millimeter-
sized drop may produce more than a hundred splinters. These ice splinters can enhance ice
initiation in convective clouds. In addition, the change in drop temperature during freezing may
exert impacts on the Hallett-Mossop process. Heymsfield and Mossop (1984) showed that the
Hallett-Mossop process is not only related to the ambient temperature but is also related to the
graupel surface temperature. In the SBM used in this study, the Hallett-Mossop process is only
parameterized for ambient temperatures between -3 °C and -8 °C. However, the Hallett-Mossop



process may occur at colder ambient temperatures if the frozen drop (or graupel) surface
temperature is appropriate (Heymsfield and Mossop, 1984). Fig. 3 shows that the drop
temperature cools from 0 °C to its ambient temperature after being fully frozen and that the
cooling rate may be even slower if there is significant riming on the surface of the particle
(Phillips et al. 2015). During this process, if the drop surface temperature and other ambient
conditions are suitable, the Hallett-Mossop process may occur at an air temperature that is colder
than -8 °C, which could also enhance the initiation of ice in developing convective clouds. For
example, a millimeter-sized frozen drop can collect approximately 600 droplets in five seconds,
assuming that the droplet concentration is 50 cm$^{-3}$ and its diameter is 20 $\mu$m. Thus, two or three
ice splinters may be produced if the ambient conditions are suitable. Moreover, time-dependent
freezing can have an impact on the dynamics in developing clouds. The instantaneous freezing of
a supercooled drop results in the sudden release of a large amount of latent heat, which may lead
to an overestimation of the vertical velocity in modeled convective clouds. In contrast, time-
dependent drop freezing can affect the cloud dynamics in a different way because its latent heat
is gradually released. Future studies are needed to explore these drop freezing-related processes.
This study reveals the importance of understanding drop freezing in convective clouds and
allows us to interpret the observed ice PSDs; however, it also raises some specific questions
about ice initiation. For example, it is not known why the observed "primary ice" concentration
is much higher than the ice nuclei concentration (DeMott et al., 2016) and the modeled ice
concentration (Fig. 1). There are several possibilities for this, including the production of ice
fragments and splinters during drop freezing or the Hallett-Mossop process; droplet collisional
freezing (Alkezweeny, 1969); or the electrofreezing of drops (Pruppacher, 1973). In addition, it
is not known whether large drops begin freezing before or after small droplets. Answering these



questions requires a better understanding of the primary drop freezing mechanisms in convective
clouds, which in turn requires more laboratory experiments to be performed and more in situ
measurements to be obtained in the future.
**4.  Summary**
In this study, the PSDs measured in fresh developing maritime convective clouds sampled during
ICE-T are shown and the deficiency of instantaneous drop freezing in models is discussed. The
observations presented here suggest that the "first ice" that is observed is small. To interpret the
observed ice PSDs, the freezing times and temperatures of supercooled drops are calculated. This
analysis indicates that the freezing time is longer for large drops than it is for small drops. Due to
the limitations of airborne instruments, freezing drops cannot be identified until they exhibit
obvious shape deformation. If the updraft is strong enough, large freezing drops may be brought
up to a colder temperature than their nucleation temperature before they begin to exhibit obvious
shape deformation. This study allows us to interpret the observed ice PSDs in fresh developing
convective clouds from the perspective of drop freezing. However, the mechanisms of drop
freezing and ice initiation are still not well known. Future studies are required to evaluate model
simulations using time-dependent drop freezing, to understand the impact of time-dependent
drop freezing on the microphysics and dynamics of convective clouds, and to further explore the
mechanisms of drop freezing and ice initiation.
**Acknowledgments**
This work is supported by the National Science Foundation (Awards AGS-1230203 and AGS-
1034858), the National Basic Research Program of China under grant no. 2013CB955802 and
the DOE Grant DE-SC0006974 as part of the ASR program. The authors acknowledge the crew



of the NCAR C-130 and the SPEC Learjet for collecting these data and providing high-quality
products. Thank Drs. Paul Lawson and Sarah Woods for processing and sharing the data of
particle size distributions measured by Learjet.



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





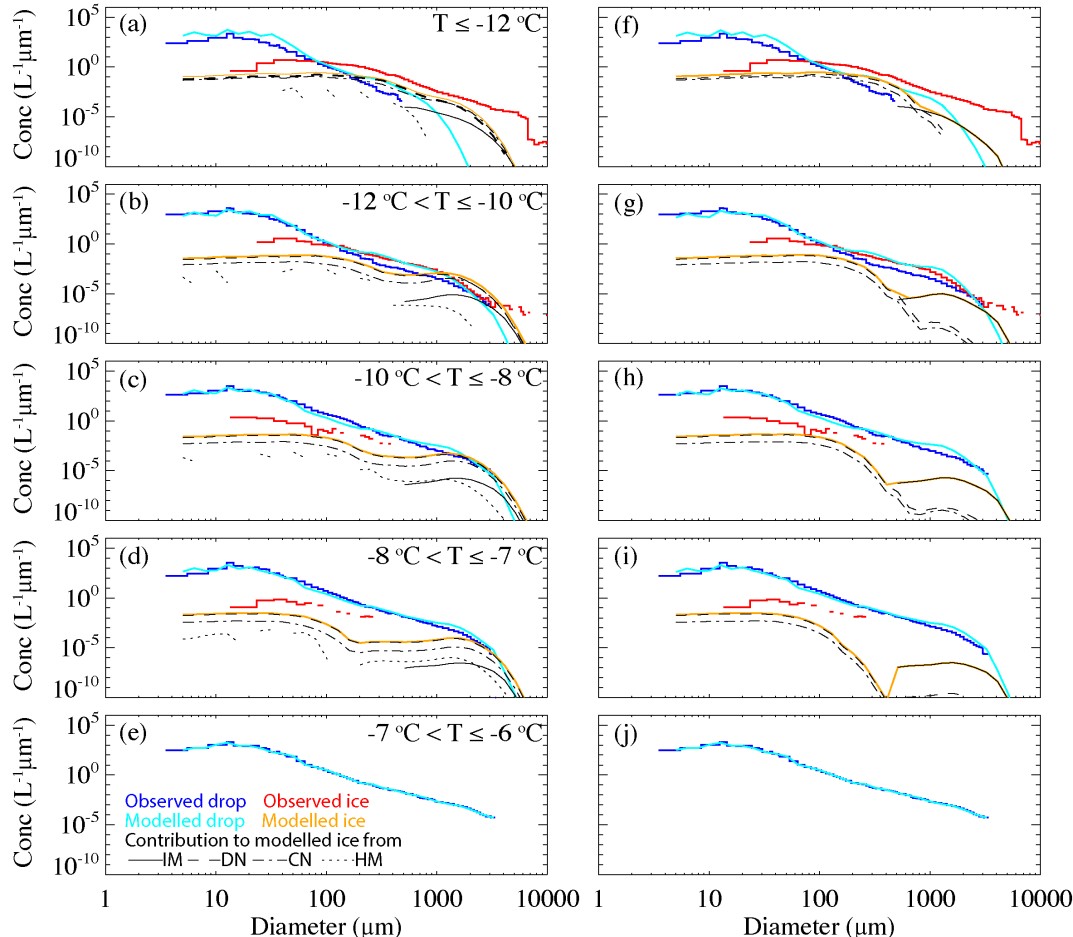

Figure 1. Particle size distributions in fresh developing convective clouds observed by the Learjet during ICE-T and those modeled using a parcel model with SBM. In the left panels, all of the ice physics implemented in the SBM are included; in the right panels, liquid-ice collision is excluded. The black solid, dashed, dashed-dotted, and dotted lines represent the contributions from immersion freezing (IM), deposition/condensation nucleation (DN), contact nucleation (CN), and the Hallett-Mossop process (HM), respectively, to the modeled ice size distributions.



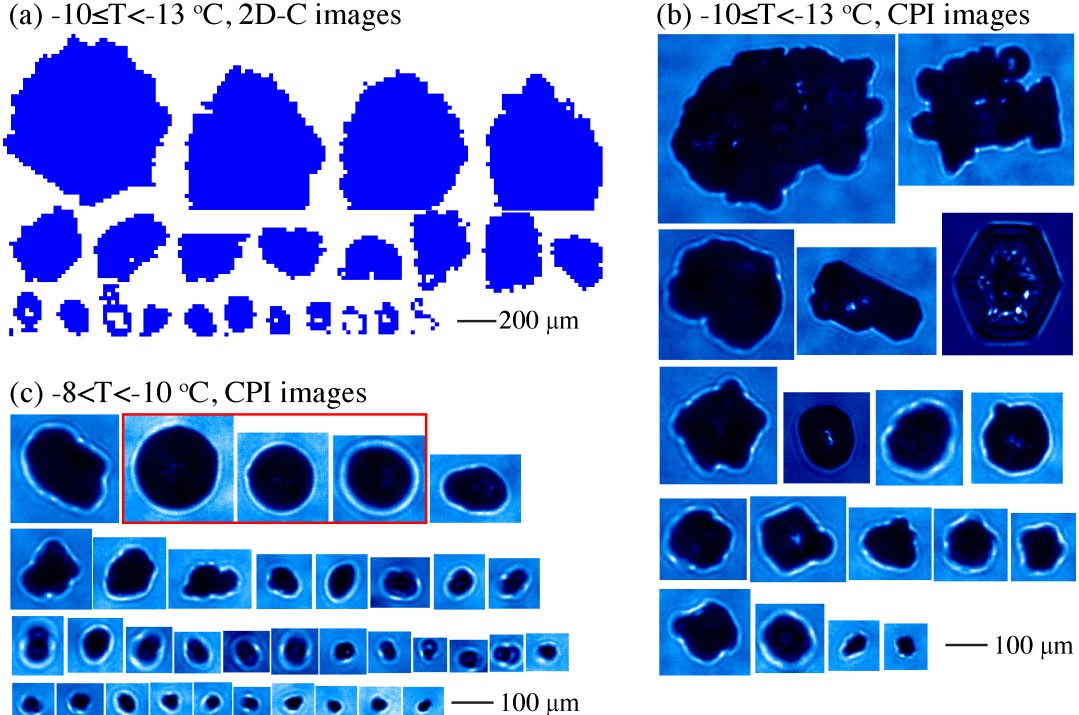

Figure 2. Examples of the 2D-C and CPI images measured in the developing convective clouds

sampled during the ICE-T project.



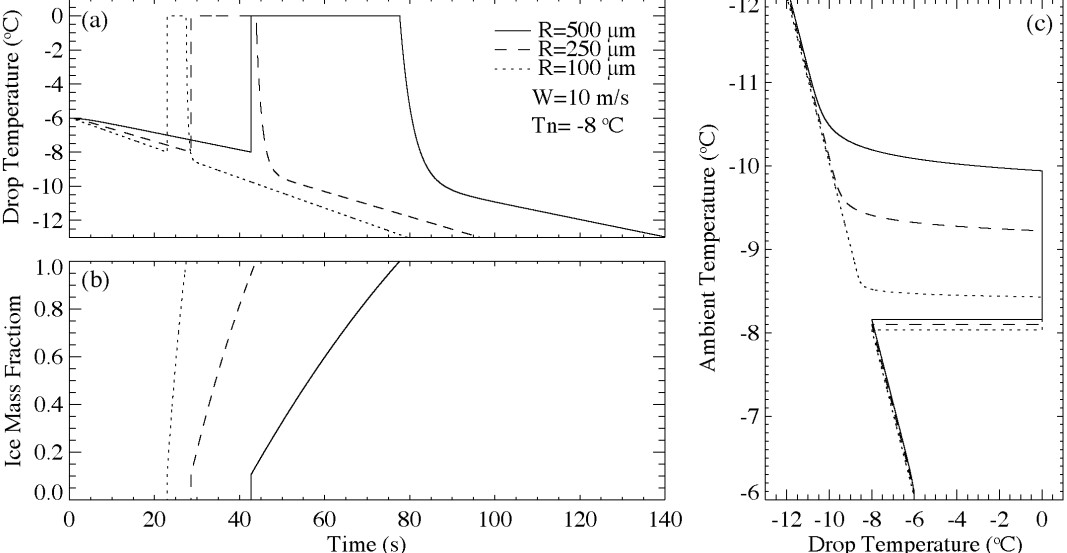

Figure 3. (a) Changes in drop temperature over time for drops with different radii. Vertical air velocity (W) is assumed to be 10 m/s and nucleation temperature (Tn) is -8 C; (b) same as (a) but for ice mass fraction; (c) ambient temperature versus drop temperature for drops with different radii.



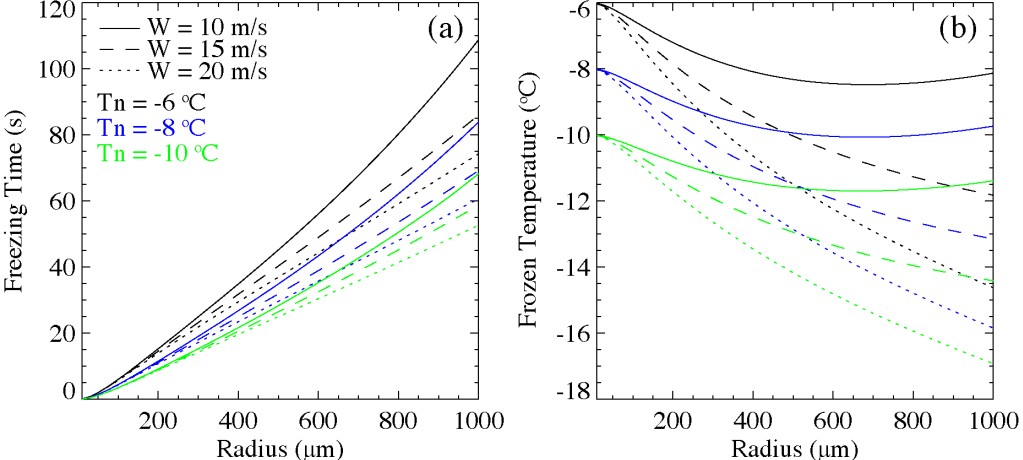

Figure 4. (a) Freezing time and (b) frozen temperature as functions of drop radius for different values of vertical air velocity (W) and nucleation temperature (Tn).



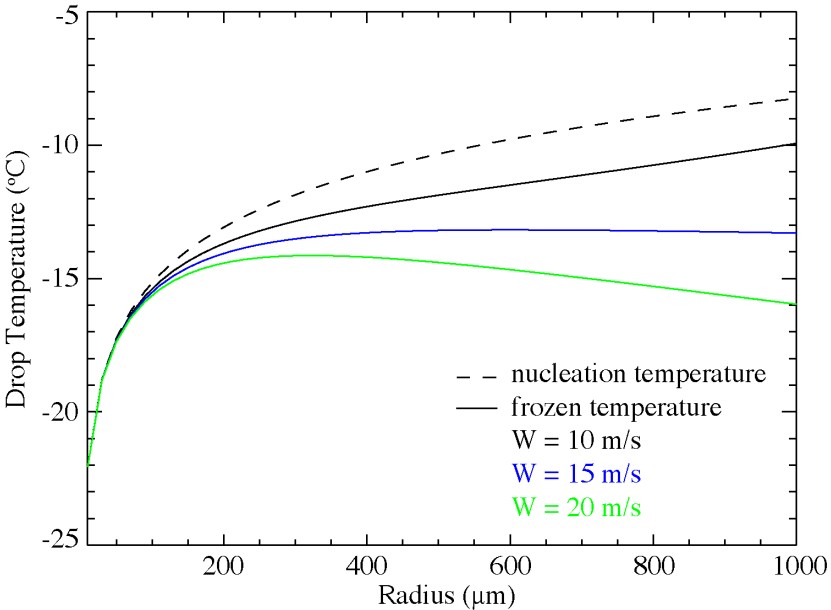

Figure 5. Drop temperature as a function of drop radius for different vertical air velocity (W)

values. The nucleation temperature is the temperature at which drops have a $10^{-4}$% probability of

freezing, as determined based on Bigg's parameterization for immersion freezing.





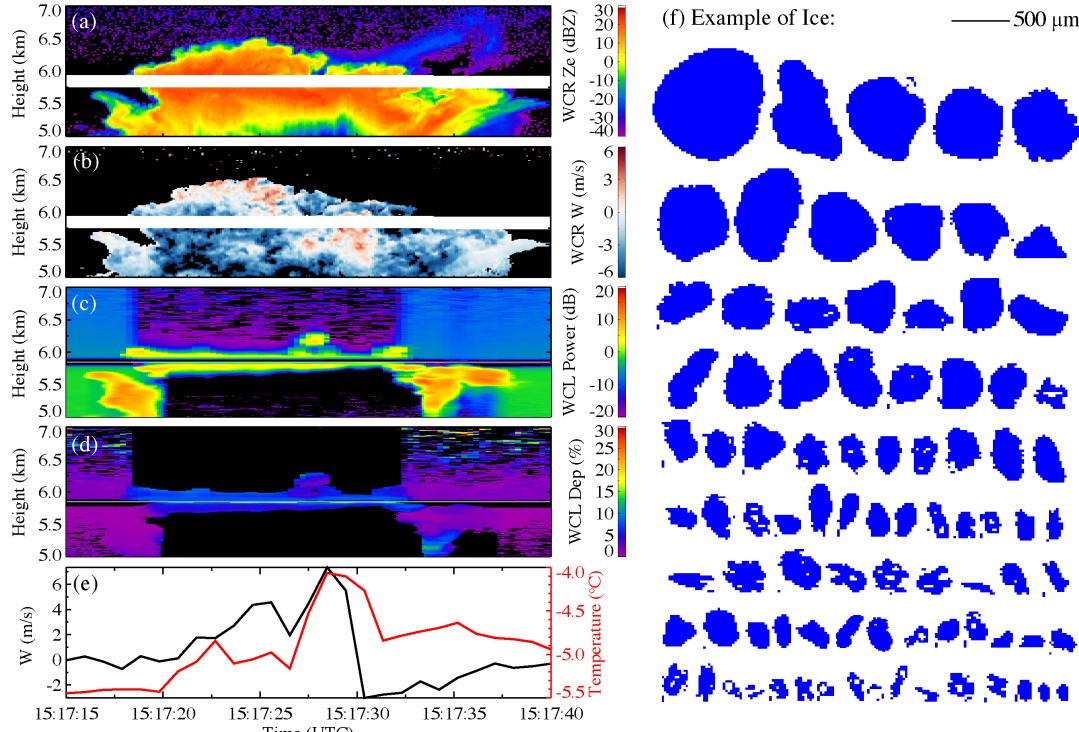

Figure 6. An example of the penetration of the C-130 in a developing cloud sampled on 23 July 2011: (a) WCR reflectivity; (b) WCR Doppler velocity; (c) WCL power; (d) WCL depolarization ratio; (e) ambient temperature and in situ vertical air velocity; and (f) examples of ice particles measured using 2D-C.