# Peer review of "On the Freezing Time of Supercooled Drops in Developing"

_Atmospheric Chemistry and Physics, 2017_

## Referee Comment (RC1) · Anonymous Referee #2 · 24 Sep 2017

In my opinion, this paper misrepresents the measurements and does not offer any new information. One of the premises stated in the paper is that (e.g. lines 236 – 237): "The ice PSD measured by the Learjet indicates that large frozen drops were observed at colder temperatures than small ice", contradicts the ICE-T measurements shown in Lawson et al. (2015). In the example in Lawson et al. (2015) Fig. 2, there is a clear progression showing that large supercooled drops freeze prior to small drops in the updraft, and the ice PSD shows much higher concentrations of small ice than large frozen drops at the coldest temperatures. These measurements are supported in the mean PSDs shown in Figs. 5 and 9, which are based on all of the cloud penetrations shown in Table 1.

The paper also contains several statements that display a lack of understanding of the

literature and the physics of convective clouds.

For example, lines 30 - 31 state: "Observations suggest that ice initiation in convective clouds is strongly related to the freezing of supercooled drops (Rangno and Hobbs, 2005; Lawson et al., 2015). While the statement has validity as written, the main conclusion of both of these studies is that ice initiation and ice production are strongly related to the size of supercooled drops.

Lines 34 - 35 state: "...during airborne measurements, freezing drops cannot be observed until they have experienced obvious deformation. This generalized statement does not apply in all situations, and perhaps does not apply in most situations. Large drops tend to deform when they freeze so that frozen drops larger than about 300 microns, which are 30 pixels across when viewed with the 2D-S, are most often discernable from supercooled drops of the same size. Smaller drops are readily discernable with the CPI, which has a much smaller sample volume and has poor sampling statistics for drops larger than about 300 microns. That said, there are certainly frozen drops that can be mistaken for supercooled drops, but laboratory experiments and comparisons with other in situ instruments (e.g., LWC and TWC devices), suggest these instances are in the minority.

Lines 39 – 40 state: "the recalescence stage [of the freezing process of a supercooled drop], during which rapid kinetic ice nucleation occurs, which results in a rapid drop in temperature that is terminated when the drop temperature reaches 0 °C." The temperature of a supercooled drop doesn't fall during the freezing process, it rises.

Line 45: Stating the (long) freezing time of a drop in still air ("typical air conditions") is of no value and misleading, because as stated later in the paper, drop cooling and freezing time is largely a function of the convective heat transfer term, which is a nonlinear function of drop size.

Lines 77 - 78: First ice in cumulus may, or may not be small. First ice can only be reliably identified using high resolution imagery from the CPI or similar instruments,
and the CPI and other similar instruments have relatively small sample volumes. This limits the ability to detect first ice, whether it is small or large. There are no conclusive measurements of the size and type of "first ice" in convective updrafts. In my opinion this is not well clarified in Lawson et al. (2015).

Line 94: The paper states that coalescence is neglected in the model. Yet, there can be no argument that this is the process that generates the large supercooled drops in tropical convective updrafts. This is a blatant oversight. This alone is grounds for rejection of the paper, unless the authors can definitively show that neglecting coalescence does not affect their results, and explain why this is so.

Lines 107 – 108: "The primary goal of the Learjet was to make repeated penetrations in fresh developing convective updrafts near the cloud top." This is incorrect. The primary objective of the Learjet was to make rapid, repeated penetrations of the updrafts of growing turrets at different altitudes. The C-130 was not capable of making rapid, repeated penetrations of growing convective turrets, and was certainly not capable of climbing with the updraft, which is what the Learjet did whenever possible.

Line 197: As stated in the paper: "The latent heat released due to freezing leads to a sudden drop in temperature from -8 °C to 0 °C." Again, this is incorrect, either a careless mistake or a fundamental misunderstanding. Going from -8 °C to 0 °C is obviously a rise in temperature.

Section 3.3 Discussion: The whole introduction of Hallett-Mossop ice multiplication is off topic and incorrect. H-M did not occur in strong ICE-T updrafts.

The only correct result from this paper that I can find is that large supercooled drops take longer to freeze than small drops. But, this has been known for hundreds of years.

I don't see how this paper can be salvaged in its present form. The work needs to be redone and the paper resubmitted.

2017.

---

## Referee Comment (RC2) · Anonymous Referee #1 · 2 Oct 2017

The authors found that the observed PSDs in fresh developing convective clouds is narrower than predicted. They argued that this is due to the fact that the freezing time for large drops is longer than that for small drops. The idea is interesting, but it is not convincing for me.

1. Based on the observational data, the "first ice" is small. But this might be due to secondary ice production or sampling statistical problem, which has been well discussed in Lawson et al. (2015). The authors did not discuss those possibilities and this would lead misleading if the readers are not familiar with this field.

2. The modeled PSD strongly depends on the ice nucleation parameterizations. By changing to another microphysical scheme, it is possible that the simulated PSD might be better or worth compared with observations. It is difficult to say whether the simulated broader PSDs is because the model does not consider the time-dependent drop freezing or the microphysical scheme.

3. In Fig. 4 and 5, the authors show results for 10, 15 20 m/s, but it is very rare that such high velocity can exist for 120 s in real clouds. For example, as shown in Fig 6, also mentioned by the authors, "the maximum updraft velocity is 7 m/s and mean updraft velocity approximately 3 m/s". It would be good to see the modeling result for low velocity. In addition, it is also useful to add the value of vertical velocity at different levels in Figure 1 a-e. I guess the velocity is smaller than 10 m/s, if so, the influence of low velocity on freezing time would be small, or even ignorable.

To sum up, the main conclusion of this paper that the observed "first ice" is due to the effect of time-dependent freezing time is not convincing for me. Secondary ice production and sampling statistics which can explain the observed "first ice" are not well discussed in this paper. I'm afraid that this would lead the effect of freezing time not-very-strong speculation or even wrong explanation. In addition, the simulation results might strongly depend on the parameterizations, which makes the comparison (to observational data) less convincing. The vertical velocity used in the model is also too strong compared with the real case, which might enhance the effect of time-dependent freezing process.

---

## Author Comment (AC1) · 10 Nov 2017

Reviewer's comments in black, replies in blue.

The authors found that the observed PSDs in fresh developing convective clouds is narrower than predicted. They argued that this is due to the fact that the freezing time for large drops is longer than that for small drops. The idea is interesting, but it is not convincing for me.

Answer: We appreciate your insightful comments and suggestions. The manuscript is revised accordingly, and is much improved based on your comments. We believe the revised manuscript is now more convincing.

1. Based on the observational data, the "first ice" is small. But this might be due to secondary ice production or sampling statistical problem, which has been well discussed in Lawson et al. (2015). The authors did not discuss those possibilities and this would lead misleading if the readers are not familiar with this field.

Answer: We appreciate the insightful comment. Ice generation in convective clouds is complicated, and we agree that some of the observed small ice at about -8 C may be due to secondary ice production. If that is the case, it means there were large drops that freeze and then produce the secondary ice. These large freezing drops may be at the early stage of freezing, so they are spherical or quasi-spherical. When these large freezing drops are not fully frozen, and contain more liquid mass than ice mass, they may be not regarded as ice based on shape. However, in the model, drop freezing is assumed instantaneous, and freezing drops are added to the ice PSD as soon as they are nucleated (please see reply to Comment 2), this is not true for large drops in real clouds at relatively warm temperatures, and thus may be a source of uncertainty in model simulations. For example, instantaneous freezing results in the sudden

release of a large amount of latent heat, while time-dependent freezing results in a gradual release of latent heat. The objective of this study is not to deal with first ice or ice generation, but focusing on understanding the freezing time for large drops and its possible consequences in interpreting observations and modelling ice generation in the models.

The sampling statistical problem is also very important in studying the ice PSDs in convective clouds. In this study, we focus on relatively warm temperature, with 14 penetrations between -7 C and -10 C, and 6 penetrations between -10 C and -12 C. The samples size is relatively small, but the observations are useful for studying ice generation in tropical maritime developing convective clouds, and can provide very useful information of the PSDs, at least for the clouds sampled by Learjet during ICE-T. Currently, there are not many measurements of PSDs in tropical maritime developing convective clouds with strong updraft cores, especially for small ice PSDs, so more field measurements are needed in the future. These points have been added to the revised manuscript.

2. The modeled PSD strongly depends on the ice nucleation parameterizations. By changing to another microphysical scheme, it is possible that the simulated PSD might be better or worse compared with observations. It is difficult to say whether the simulated broader PSDs is because the model does not consider the time-dependent drop freezing or the microphysical scheme.

Answer: We totally agree that the modelled PSD strongly depends on the ice nucleation parameterizations, and we apologize for the unclear discussion on this point in the original manuscript. To confirm that the modelled broader PSD is because time-dependent drop freezing is not considered, we conducted two simulations, as seen in Fig. R1 below, in the left panels, all of the ice physics implemented in the SBM are included; in the right panels, liquid-ice collision

is excluded. If liquid-ice collision is excluded, most of the modelled ice is small at warm temperatures, consistent with observation (Fig. 1h and i), only a very low concentration of large ice is found, which is from immersion freezing. If liquid-ice collision is included, the concentration of large ice suddenly increases by 2 orders of magnitude at temperature between -7 C and -10 C (Fig. 1c and d), indicating these large ice are from liquid-ice collision process, and are added to the ice PSD as soon as they are nucleated in the bin microphysical scheme we used, which is widely use for WRF simulations. Taking into account the long freezing time of large drops, these particles actually are at the freezing stage, but not fully frozen. This discussion has been added to the revised manuscript.

3. In Fig. 4 and 5, the authors show results for 10, 15 20 m/s, but it is very rare that such high velocity can exist for 120 s in real clouds. For example, as shown in Fig 6, also mentioned by the authors, "the maximum updraft velocity is 7 m/s and mean updraft velocity approximately 3 m/s". It would be good to see the modeling result for low velocity. In addition, it is also useful to add the value of vertical velocity at different levels in Figure 1 a-e. I guess the velocity is smaller than 10 m/s, if so, the influence of low velocity on freezing time would be small, or even ignorable.

Answer: Great idea! We have added the observed mean vertical velocity and averaged maximum vertical velocity in Fig. 1, please see Fig. R1 below. We also modified Fig. 3, 4 and 5 based on the observed vertical velocity, please see Figs. R3, R4, R5 below. As seen in Fig. R1, the observed mean vertical velocity is approximately 10 m/s between -6 C and -10 C, the averaged maximum vertical velocity is approximately 15 m/s between -6 C and -10 C. The strong updraft at temperatures warmer than -10 C can transport large supercooled drops to higher

levels. For example, in Fig. R3, if a supercooled drop in 500 microns in radii starts freezing at about -8 C, it is fully frozen at about -10.18 C. This is also confirmed in Fig. R4c and d, due to the relatively long freezing time and strong updraft, drops larger than 400 microns in radii which start freezing at -8 C or -6 C are fully frozen at temperatures 2–3 C colder than the nucleation temperature. At temperatures colder than -10 C, the observed updrafts are weaker, so large supercooled drops can stay at the same level for a long time, and the freezing temperature is similar to the nucleation temperature.

We totally agree that if the vertical velocity is weaker, the influence of vertical velocity on the freezing temperature would be smaller. The purpose of Fig. 6 is to support this point. In Fig. 6, the cloud has a relatively weak updraft, and large particles may stay at the same level for a long time or may fall through the updraft from aloft, so graupel and frozen drops were observed at temperatures as warm as -5 C, this is different than Fig. 1, which is plotted for much stronger updrafts. In another study (Yang et al. 2016), we plot the PSDs for all the updrafts stronger than 1 m/s sampled by C-130 during ICE-T, the results also support that large ice are observed at warm temperatures in developing convective clouds with relatively weak updraft.

*Reference:*

*Yang, J., Wang, Z., Heymsfield, A. J., and Luo, T.: Liquid-Ice Mass Partition in Tropical Maritime Convective Clouds. J. Atmos. Sci., 73, 4959-4978, 2016.*

[Figure]

Figure R1. Particle size distributions in fresh developing convective clouds observed by the Learjet during ICE-T and those modeled using a parcel model with SBM. In (a)-(e), all of the ice physics implemented in the SBM are included; in (f)-(j), drop-ice collision is excluded. The black solid, dashed, and dashed-dotted lines in (f)-(j) represent the contributions from immersion freezing (IM), deposition/condensation nucleation (DN), and contact nucleation (CN), respectively, to the modeled ice size distributions. The black solid and dashed-dotted lines in (a)-(e) represent the contributions from primary ice nucleation (IM+DN+CN) and drop-ice collision, respectively, to the modeled ice size distributions. The observed mean vertical velocity ($W_{mean}$) and averaged maximum vertical velocity ($W_{max}$) are shown in (a)-(e).

[Figure]

Figure R3. (a) Changes in drop temperature over time for drops with different radii based on the observed mean vertical velocity, which is temperature-dependent. Nucleation temperature (Tn) is -8 °C; (b) same as (a) but for ice mass fraction; (c) ambient temperature versus drop temperature for drops with different radii. The red solid, dashed and dotted lines indicate the frozen temperature for drops with radius of 500 μm, 250 μm and 100 μm, respectively.

[Figure]

Figure R4. (a) Freezing time and (b) frozen temperature as functions of drop radius for different values of vertical air velocity (W) and nucleation temperature (Tn). (c) and (d) are the same as (a) and (b) but for the observed mean vertical velocity ($W_{mean}$) and averaged maximum vertical velocity ($W_{max}$), which are temperature-dependent.

[Figure]

Figure R5. Drop temperature as a function of drop radius for different vertical air velocity (W) values, including the observed mean vertical velocity ($W_{mean}$) and averaged maximum vertical velocity ($W_{max}$), which are temperature-dependent. The nucleation temperature is the temperature at which drops have a $10^{-4}\%$ probability of freezing, as determined based on Bigg's parameterization for immersion freezing.

To sum up, the main conclusion of this paper that the observed "first ice" is due to the effect of time-dependent freezing time is not convincing for me. Secondary ice production and sampling statistics which can explain the observed "first ice" are not well discussed in this paper. I'm afraid that this would lead the effect of freezing time not-very-strong speculation or even wrong explanation. In addition, the simulation results might strongly depend on the parameterizations, which makes the comparison (to observational data) less convincing. The

vertical velocity used in the model is also too strong compared with the real case, which might enhance the effect of time-dependent freezing process.

Answer: We appreciate your insightful comments and suggestions. The manuscript is revised accordingly. The objective of the paper is not to focus on "first ice" or ice generation but to focus on understanding the freezing time of large drops and its possible consequences in interpreting observations and modeling ice generation in the models. Discussion about secondary ice production and sampling statistics are added. The impact of ice microphysics in the model simulations is better discussed to confirm that the modelled broad ice PSD is due to the instantaneous drop freezing. Fig. 1, 3, 4, and 5 are modified based on the observed vertical velocity, and text is revised accordingly. The revised manuscript is much improved based on these comments, and we believe the results is more convincing now.

---

## Author Comment (AC2) · 10 Nov 2017

The comment was uploaded in the form of a supplement: https://www.atmos-chem-phys-discuss.net/acp-2017-714/acp-2017-714-AC2-supplement.pdf

---

## Author Comment (AC3) · 10 Nov 2017

Reviewer's comments in black, replies in blue.

In my opinion, this paper misrepresents the measurements and does not offer any new information. One of the premises stated in the paper is that (e.g. lines 236 – 237): "The ice PSD measured by the Learjet indicates that large frozen drops were observed at colder temperatures than small ice", contradicts the ICE-T measurements shown in Lawson et al. (2015). In the example in Lawson et al. (2015) Fig. 2, there is a clear progression showing that large supercooled drops freeze prior to small drops in the updraft, and the ice PSD shows much higher concentrations of small ice than large frozen drops at the coldest temperatures. These measurements are supported in the mean PSDs shown in Figs. 5 and 9, which are based on all of the cloud penetrations shown in Table 1.

Answer: We appreciate your careful reading of the manuscript and your insightful comments. The purpose of this study is to quantitatively estimate the freezing time of supercooled drops with different sizes, and to show the importance of the drop freezing time in understanding aircraft observations, modeling drop freezing, and evaluating models using aircraft measurements, which are rarely discussed in previous studies. This is important because in models freezing of supercooled drops are assumed instantaneous; this is not true and is an oversight in developing models. Assuming instantaneous freezing may be an important source of uncertainties in modelling convective clouds, for example, instantaneous freezing of supercooled drops results in the sudden release of a large amount of latent heat, which may lead to an overestimation of the vertical velocity in modeled convective clouds.

According to Fig 2, 5 and 9 in Lawson et al. (2015), you suggest that "*large supercooled drops freeze prior to small drops in the updraft, and the ice PSD shows much higher*

*concentrations of small ice than large frozen drops at the coldest temperatures*". This does not contradict our study, and we agree that large supercooled drops are likely to start to freeze prior to small drops in the updrafts. As seen in Fig. 2 and 5 in Lawson et al. (2015), the ice observed at about -8 C were small, with larger frozen drops observed at temperatures colder than -8 C. Taking into account the relatively long freezing time for large drops, the observed frozen drops may start freezing at temperatures warmer than observed within the strong updraft core. We are not arguing the drops at the early stage of freezing should be regarded as ice, actually these drops probably contain more liquid mass than ice mass. However, in models any supercooled drop that begins to freeze is regarded as a fully frozen drop; this is why there is large ice at about -8 C in the simulation, inconsistent with airborne observations (Fig. 1). In the revised manuscript *"The ice PSD measured by the Learjet indicates that large frozen drops were observed at colder temperatures than small ice"* is changed to "*The ice PSD measured by the Learjet indicates that ice observed at about -8 C are primarily small, and the larger frozen drops were observed at temperatures colder than -8 C*".

The paper also contains several statements that display a lack of understanding of the literature and the physics of convective clouds.

For example, lines 30 – 31 state: "Observations suggest that ice initiation in convective clouds is strongly related to the freezing of supercooled drops (Rangno and Hobbs, 2005; Lawson et al., 2015). While the statement has validity as written, the main conclusion of both of these studies is that ice initiation and ice production are strongly related to the size of supercooled drops.

Answer: We appreciate the comment. *"Observations suggest that ice initiation in convective clouds is strongly related to the freezing of supercooled drops"* is changed to *"Observations suggest that ice initiation in convective clouds is strongly related to the freezing of supercooled drops and the size of the freezing drops"*.

Lines 34 - 35 state: "... during airborne measurements, freezing drops cannot be observed until they have experienced obvious deformation. This generalized statement does not apply in all situations, and perhaps does not apply in most situations. Large drops tend to deform when they freeze so that frozen drops larger than about 300 microns, which are 30 pixels across when viewed with the 2D-S, are most often discernable from supercooled drops of the same size. Smaller drops are readily discernable with the CPI, which has a much smaller sample volume and has poor sampling statistics for drops larger than about 300 microns. That said, there are certainly frozen drops that can be mistaken for supercooled drops, but laboratory experiments and comparisons with other in situ instruments (e.g., LWC and TWC devices), suggest these instances are in the minority.

Answer: We apologize for the unclear statement. We agree large drops tend to deform when they are close to fully frozen, and frozen drops larger than 300 microns are mostly discernable from supercooled drops using 2D-S images, small frozen drops are readily discernable with the CPI. However, it takes time for the drops to freeze and thus deform, so for drops at the early stage of freezing the deformation is slight, so they are generally spherical. In the revised manuscript, *"... during airborne measurements, freezing drops cannot be observed until they have experienced obvious deformation"* is changed to *"drops at the early stage of freezing usually has no or slight deformation"*.

Lines 39 – 40 state: "the recalescence stage [of the freezing process of a supercooled drop], during which rapid kinetic ice nucleation occurs, which results in a rapid drop in temperature that is terminated when the drop temperature reaches 0 C." The temperature of a supercooled drop doesn't fall during the freezing process, it rises.

Answer: We apologize for the mistake, "...*which results in a rapid drop in temperature...*" is now changed to "...*which results in a rapid rise in temperature...*".

Line 45: Stating the (long) freezing time of a drop in still air ("typical air conditions") is of no value and misleading, because as stated later in the paper, drop cooling and freezing time is largely a function of the convective heat transfer term, which is a nonlinear function of drop size.

Answer: Thank you for the comment. The sentence: "*In typical air conditions, it takes approximately 400 s for a stationary millimeter-sized drop to completely freeze at -5 °C under a pressure of 1 atm; this freezing time is reduced to approximately 200 s at -10 °C*" is removed in the revised manuscript.

Lines 77 – 78: First ice in cumulus may, or may not be small. First ice can only be reliably identified using high resolution imagery from the CPI or similar instruments, and the CPI and other similar instruments have relatively small sample volumes. This limits the ability to detect first ice, whether it is small or large. There are no conclusive measurements of the size and type of "first ice" in convective updrafts. In my opinion this is not well clarified in Lawson et al. (2015).

Answer: We agree there are no conclusive measurements of the size and type of "first ice" in convective updrafts. In the revised manuscript, "*first ice*" is changed to "*ice at relatively warm temperature (about -8 C)*". *Similar statements elsewhere have* been clarified too.

Line 94: The paper states that coalescence is neglected in the model. Yet, there can be no argument that this is the process that generates the large supercooled drops in tropical convective updrafts. This is a blatant oversight. This alone is grounds for rejection of the paper, unless the authors can definitively show that neglecting coalescence does not affect their results, and explain why this is so.

Answer: We totally agree that coalescence is the key process that generates large supercooled drops in convective clouds. Firstly, we'd like to clarify that coalescence is included in the parcel model to model cloud microphysics, but we neglect the impact of coalescence on the calculation of the freezing time of a single supercooled drop. It is not necessary to include the coalescence process in the calculation of the freezing time of a single supercooled drop, because the freezing time is the shortest assuming that drop size doesn't increase due to coalescence after freezing begins. The drop size increase due to the coalescence process results in a longer freezing time. In this study, we prefer to examine the shortest freezing time of supercooled drop for a given size. The results indicate that the freezing time of supercooled drops larger than 200 microns in diameter is considerably long even if size increase due to coalescence is not included (longer than the typical time step of cloud resolving models), therefore, freezing of supercooled drop is an important process which should be considered in numerical models, but few models have done so.

Lines 107 – 108: "The primary goal of the Learjet was to make repeated penetrations in fresh developing convective updrafts near the cloud top." This is incorrect. The primary objective of the Learjet was to make rapid, repeated penetrations of the updrafts of growing turrets at different altitudes. The C-130 was not capable of making rapid, repeated penetrations of growing convective turrets, and was certainly not capable of climbing with the updraft, which is what the Learjet did whenever possible.

Answer: We appreciate for the correction. *"The primary goal of the Learjet was to make repeated penetrations in fresh developing convective updrafts near the cloud top"* is changed to *"The primary objective of the Learjet was to make rapid, repeated penetrations in the updrafts of growing turrets"*.

Line 197: As stated in the paper: "The latent heat released due to freezing leads to a sudden drop in temperature from -8 C to 0 C." Again, this is incorrect, either a mistake or a fundamental misunderstanding. Going from -8 C to 0 C is obviously a rise in temperature.

Answer:

We apologize for the mistake, *"The latent heat released due to freezing leads to a sudden drop in temperature from -8 C to 0 C"* is now changed to *"The latent heat released due to freezing leads to a sudden rise in temperature from -8 C to 0 C"*.

Section 3.3 Discussion: The whole introduction of Hallett-Mossop ice multiplication is off topic and incorrect. H-M did not occur in strong ICE-T updrafts.

Answer: Thank you for the comment, we agree that the H-M process did not occur in strong developing ICE-T updrafts. The discussion about H-M process is removed in the revised manuscript.

The only correct result from this paper that I can find is that large supercooled drops take longer to freeze than small drops. But, this has been known for hundreds of years. I don't see how this paper can be salvaged in its present form. The work needs to be redone and the paper resubmitted.

Answer: We acknowledge your comments and suggestions. The purpose of this study is to show the importance of the freezing time to more fully understand aircraft observations, to model drop freezing, to evaluating models using aircraft measurements, and to note the deficiency of instantaneous drop freezing currently assumed in cloud models. The theory of heat transfer in the calculation of the freezing time of supercooled drops is not new, but the importance of considering the freezing time of large supercooled drops in modeling ice PSDs, in understanding the observed ice PSDs, and effectively using observations to validate and improve simulations has not been appreciated. We try to correct that oversight in our article.

We apologize for the unclear statements. The manuscript has been much improved based on your comments, and unclear statements have been clarified. For example, statements like "*freezing drops cannot be observed until they have obvious shape deformation*" are changed to "*drops at the early stage of freezing usually have a slight deformation*". Please see more revisions in the revised manuscript with track changes.